# Comprehensive Analysis of the Structure and Allergenicity Changes of Seafood Allergens Induced by Non-Thermal Processing: A Review

**DOI:** 10.3390/molecules27185857

**Published:** 2022-09-09

**Authors:** Fengqi Wang, Hangyu Zhong, Jun-Hu Cheng

**Affiliations:** 1School of Food Science and Engineering, South China University of Technology, Guangzhou 510641, China; 2Academy of Contemporary Food Engineering, South China University of Technology, Guangzhou 510006, China

**Keywords:** non-thermal processing technologies, seafood allergen, structure, allergenicity

## Abstract

Seafood allergy, mainly induced by fish, shrimp, crab, and shellfish, is a food safety problem worldwide. The non-thermal processing technology provides a new method in reducing seafood allergenicity. Based on the structural and antigenic properties of allergenic proteins, this review introduces current methods for a comprehensive analysis of the allergenicity changes of seafood allergens induced by non-thermal processing. The IgE-binding capacities/immunoreactivity of seafood allergens are reduced by the loss of conformation during non-thermal processing. Concretely, the destruction of native structure includes degradation, aggregation, uncoiling, unfolding, folding, and exposure, leading to masking of the epitopes. Moreover, most studies rely on IgE-mediated assays to evaluate the allergenic potential of seafood protein. This is not convincing enough to assess the effect of novel food processing techniques. Thus, further studies must be conducted with functional assays, in vivo assays, animal trials, simulated digestion, and intestinal microflora to strengthen the evidence. It also enables us to better identify the effects of non-thermal processing treatment, which would help further analyze its mechanism.

## 1. Introduction

Seafood, including fish, shrimp, crab, squid, algae, shellfish, and others [1], are popular for their palatability and great nutritional value, such as protein, vitamins, calcium, and irons. Furthermore, eating seafood generates various positive effects, such as antioxidant, anti-cancer, regulation of blood sugar, and others [2,3,4]. However, most of the seafoods (e.g., shrimp, fish, crab, and squid) are known as highly allergenic foodstuffs, likely to trigger allergic symptoms (including asthma, rhinitis, abdominal cramps, vomiting, itchy skin, and even anaphylactic shock in severe cases) upon ingestion [5,6]. As such, the allergic threat of seafood is an obstacle to developing the seafood industry [7]. Recently, the number of food-allergy cases has been continuously rising [8,9], and among them, seafood allergy is relatively severe, especially in Asian countries [10].

For consumers, the immunoglobulin (IgE)-mediated type of food allergy occurs most frequently [11]. When the patients are first exposed to the allergen, their plasma cells would produce specific IgE. Then, the antibodies would attach to the fragment crystallizable receptors (FcR), which are present on the mast cells. Subsequently, mast cells would release mediators to trigger numerous allergic reactions after regaining exposure to the same allergen [12]. So far, however, the most effective therapy for food-allergy patients seems to be avoiding ingesting them. Therefore, how to reduce or eliminate the potential sensitization capacities of seafood has become a hot topic.

Traditionally, heat treatment is a common method to tackle allergic food problems, because their structure generally would be destroyed in high temperature, and even denatured. However, due to the heat stability of some seafood allergens (e.g., tropomyosin and parvalbumin), thermal treatment cannot make a huge difference in their allergenicity [13]. Sometimes, heat treatment even causes an increase in allergenicity [14]. Although extreme thermal conditions may obtain effective results [15], they could result in irreversible losses in food sensory properties and nutritional value. Moreover, chemical and biological methods are also ineffective and uncontrollable in reducing allergenic potential of seafood allergens [16,17,18]. Therefore, novel non-thermal processing technologies are urgently required to solve the challenge of seafood allergies.

Non-thermal processing technologies, including high pressure processing (HPP), cold plasma (CP), ultrasound, pulsed electric field (PEF), and ultraviolet (UV), are considered as effective solutions to food allergens [19,20]. It was reported that side effects of some food allergens were largely reduced under non-thermal processing treatment [21,22,23,24,25,26]. Moreover, new processing technologies also demonstrate their unique effects on the heat-resistance characteristics of seafood allergens. For example, Jin et al. [27] used HPP to process the main allergens of squid (*Todarodes pacificus*) and found that its immunoreactivity decreased. Ekezie et al. [28] found that the CP jet had a good reducing effect on the IgE-binding capacity of the main allergen of shrimp (*Litopenaeus vannamei*). Shriver et al. [29] also found that UV could change the IgE/IgG-binding capacity of shrimp (*Litopenaeus setiferus*) allergens. Furthermore, the non-thermal treatment also has other modificatory advantages on seafood, such as improving the gel properties of surimi [30,31], increasing the hardness of shrimp [32], enhancing the efficacy of active substances [33], and prolonging the shelf life [34]. Therefore, non-thermal processing technology can provide a new possibility for reduction in seafood allergenicity.

The structure of allergens is an important factor that conserves its allergenic potency. For instance, the tropomyosins of crustaceans present a visible α-helices structure, while the arginine kinase of crab and parvalbumin of fish show a compact globular structure. Their integrity of unique structure is always related to their allergenicity. In the studies of seafood allergens treated by HPP, researchers mainly focused on the changes in secondary structure and tertiary structure of allergenic proteins, and thus inferred the mechanism about reducing allergenicity from the structural changes [35,36,37]. For example, Jin et al. [27] suggested that HPP might reduce the immunoreactivity of the tropomyosin of squid by expanding and modifying its secondary structure (53% α-helix was converted into β-sheet and random coils at 600 MPa for 20 min). Therefore, the relationship between the structure and allergenicity of seafood allergens is a crucial way to explore the mechanism of novel processing techniques. As non-thermal processing techniques have a variety of physicochemical effects, the analysis and impacts of structural changes of seafood allergens after non-thermal processing treatment are not comprehensive enough. It is still unclear how the protein structure unfolds or folds during the treatment. In addition, the approaches to access the impact on allergenicity still rely on an IgE/IgG-mediated assay. It is an acceptable but not comprehensive evaluation on allergenicity. The measures of the allergenicity of seafood after treatment require higher weight of evidence (WOE) determinations for verification.

Therefore, focusing on the insights into seafood allergens treated by non-thermal processing and the weakness of structure-allergenicity analysis, this review summarizes the modificatory effect on structure and allergenicity of seafood allergens after non-thermal processing treatment. In addition, new methods of comprehensive analysis are concluded in this review (shown in Figure 1). It would help to establish the relationship between structure and allergenicity, which is of great significance for thoroughly understanding the follow-up study about treating the seafood allergen.

## 2. Analysis of Structure Changes of Seafood Allergens Induced by Non-Thermal Processing

Most non-thermal processing techniques would break the non-covalent bonds of proteins, leading to the destruction of structure and conformation. Previous studies showed that the immunoreactivity would be influenced once the processing modified the structure of proteins [38,39]. The vast majority of seafood allergens are proteins. Commonly, seafood allergens include tropomyosin (e.g., shrimp, squid, crab, abalone, and fish), parvalbumin (e.g., fish), arginine kinase (e.g., crayfish and lobster), and hemocyanin (e.g., octopus and scallop) (http://allergen.org, accessed on 1 March 2022). In essence, proteins are polypeptide chains formed by the covalent linkage of amino acids, and each molecule has its unique three-dimensional structure. Therefore, the structure of proteins is generally divided into the primary structure, secondary structure, tertiary structure, and quaternary structure [40]. The mechanism of allergen-antibody binding is closely related to its linear epitopes and conformational epitopes [41]. Between them, linear epitopes are peptides composed of 10~20 consecutive amino acids on the amino acid sequence, while conformational epitopes are regions formed of 4~6 amino acids in space [42]. There is a closed relationship between structure and epitopes [43]. Undeniably, non-thermal processing usually results in structural changes of the allergen, leading to masking, exposure, or destruction of its epitopes (shown in Table 1) [19]. Consequently, it is necessary to detect the structural properties of allergens after non-thermal processing treatment.

### 2.1. Primary Structure Changes

The primary structure, namely the sequence of amino acids, is closely related to the epitopes. During non-thermal processing, epitopes would be altered if the primary structure of the allergen is changed, such as chemical modification on amino acid side chains or breakage of peptide bond. As a result, the epitope would be changed, whether the linear or conformational epitope, so that the IgE-binding capacities are influenced by processing techniques [52]. Accordingly, detecting the primary structure of proteins is an effective and direct method to characterize their allergenic potential. In the studies of food allergens, two methods are usually applied to detect it. One is immediate and accurate, namely mass spectrometry (MS). It can help to detect the weight of molecules and identify the sequence of amino acids. In this way, the chemical modification on protein induced by non-thermal processing can be analyzed [53]. For example, Liu et al. [54] reported that the IgG-binding capacity (determined by ELISA and Western Blotting) of food allergen was altered up to 58.21% during 4 min dielectric-barrier discharge (DBD) plasma treatment. This result may be attributed to the oxidation of some amino acids closed to Met, Tyr, and Phe, which MS detected. However, the equipment is expensive and the process is complex. The other one is sodium dodecyl sulfate polyacrylamide gel electrophoresis (SDS-PAGE), which is frequently used to visualize the protein profile [55]. During the SDS-PAGE processing, proteins are negatively charged and move directionally in an electric field. Because of the different mobility caused by the molecular weight, proteins with different molecular weights are separated. In this way, it can reflect the change in the protein’s total molecular weight after non-thermal techniques treatment. Although it cannot specifically reflect amino sequence changes, the molecular weight can help infer whether the covalent bond breaks or the protein is degraded during processing.

In the research on non-thermal processing of seafood allergens, SDS-PAGE is used to analyze the purity of allergenic protein or confirm the presence of allergenic protein. Moreover, it can help to determine whether the allergenic protein degraded or aggregated during non-thermal processing [24]. This can be used to predict whether the polypeptide chains have broken. Furthermore, it has been found that non-thermal processing treatment could reduce the intensities of target bands of allergenic proteins, indicating that the allergen may be degraded into small molecules during the processing. For this reason, their primary structure was destroyed, leading to a decrease in IgE-binding capacities [11,28,48,51]. These results also revealed that non-thermal processing techniques could destroy peptide bonds due to the high-pressure extrusion, oxidation, etching of high-energy electrons, cavitation, and other factors. Besides the decrease in band intensities, some results also showed the appearance of new bands, such as a new band at 26 kDa of fish proteins occurring after HPP treatment [44]. This may be attributed to the low handling intensity (200 MPa, 20 min). However, due to the low sensibility, the results of electrophoresis in some time cannot completely and accurately reflect the changes in behaviors of proteins (e.g., degradation or aggregation). For instance, Zhang et al. [39] used high-intensity ultrasound to process the allergen of shrimp. The electrophoresis results merely suggested the decline of the target band intensity, but not the production of other new bands. Contradictorily, the high-performance liquid chromatography (HPLC) results showed the appearance of another fragment. This may be attributed to the large fragments which cannot pass through the pores of the polyacrylamide or the unmatched polyacrylamide concentration of the gel that cannot catch the small fragments [27,56]. In sum, allergens might degrade during the novel processing according to the SDS-PAGE detections.

### 2.2. Secondary Structure Changes

The secondary structure of a protein describes the spatial winding and folding of peptide chains. It is maintained primarily by hydrogen bonds, in the form of α-helix, β-sheet, β-turns, and random coils [57]. Changes in secondary structure may lead to alterations in linear or conformational epitopes of allergens, thus decreasing allergenicity. Furthermore, the secondary structure is also the basis of the tertiary structure; if it changes, it could cause the conformational structure to collapse [58]. Therefore, the determination of secondary structure of the allergens is also important in studies of allergens. At present, the practical detections are circular dichroism (CD) and Fourier transform infrared (FTIR) [59]. The secondary structure has different characteristic absorption peaks in the spectrum of CD and FTIR. Moreover, CD has more restrictions on the sample, including functional group, sample state, molecular weight, and concentration of protein, while FTIR does not. For example, Ekezie et al. [28] applied CD to analyze tropomyosin of king prawn (*Litopenaeus vannamei*) after CP jet treatment. The results showed that the native allergen was an ordered structure dominated by α-helix (negative absorption peak at 208 and 222 nm), and the intensity of the characteristic peak of α-helix decreased gradually after CP treatment. It was indicated that the secondary structure of tropomyosin changed from α-helix to β-sheet and β-turns (peak area at 180~200 nm increases). This may be due to the oxidative modification of amino acid by free radicals produced by CP treatment, which altered the interaction between protein molecules [53]. As the structure was changed, the IgE-binding capacities of allergens were reduced. Jin et al. [27] treated tropomyosin of squid with HPP and Zhang et al. [39] treated tropomyosin of shrimp with high-intensity ultrasound. They obtained similar results on secondary structure of protein. This may indicate that both HPP and ultrasound broke the inner hydrogen bond of tropomyosin, thus changing the protein molecular structure and affecting its allergenicity. Particularly, the attenuated immune-response of tropomyosin from seafood is often accompanied by a transition from α-helix to β-sheet and random coil [27,28,39,48,51]. In FTIR, the peak of amide I (1600–1700 cm^−1^) is the best characteristic for the secondary structure of proteins. Zhang et al. [11] found that the intensity of amide I peak in hemocyanin of squid decreased after HPP treatment. It was found that with the increase in pressure to 600 MPa, α-helix content decreased from 32.37% to 23.67%, β-turns content increased from 16.80% to 21.95%, and random coils content increased from 32.02% to 37.54%, which was similar with the report by Dong et al. [48], who applied high-intensity ultrasound to processing proteins of shrimp. With the extension of treatment time, the secondary structure changed more and more significantly. Zhang et al. [39] also found that the transformational degree of secondary structure was positively correlated with ultrasonic power. Hence, the secondary structure of seafood allergens would be markedly influenced after non-thermal treatment, showing the transition from order to disorder.

### 2.3. Tertiary Structure Changes

The tertiary structure of a protein is the conformation formed by the interaction of the amino acid side chain on the basis of the secondary structure. It mainly relies on secondary bonds, including hydrogen bonds, hydrophobic bonds, and van der Waals forces. Some of them also rely on disulfide bonds to maintain the conformation. The tertiary structure of a protein allows interaction between amino acids that are far apart in the primary structure, which may constitute conformational epitopes of allergens. Therefore, the change of tertiary structure may lead to the destruction of conformational epitopes and thus affect allergenic potential. Fluorescence chromatography is frequently used to characterize the changes in polarity of proteins. Due to their phenyl, Trp, Tyr, and Phe have fluorescent properties [60]. Hence, when the protein’s tertiary structure changes, its polarity would alter because of the phenyl exposure. This may result in a blue or red shift of the maximum emission wavelength of the fluorescence spectrum [61]. Ma et al. [50] treated cod protein with ultrasound. They found that with the increase in ultrasonic power, the inherent fluorescence intensity gradually increased, while the maximum emission wavelength (λ_max_) blue shifted from 347.5 nm to 344 nm. This result indicated that the chromogenic groups of cod protein were exposed more in the aqueous phase after ultrasonic treatment, increasing the hydrophobicity (non-polarity) inside the protein. In other words, the tertiary structure of the cod protein was destroyed, and the structure unfolded after non-thermal processing. Conversely, another research found that the fluorescence intensity of shrimp allergen (tropomyosin) treated with CP decreased by about 16%, while λ_max_ red shifted from 338 nm to 339 nm [28]. This may be because the CP treatment decreased its particle size, making aggregation or recombination of protein easier. This aggregating behavior would mask the internal chromogenic groups, reducing the number of groups exposed to the aqueous phase. Zhang et al. [45] obtained similar results on the fluorescence intrinsic spectrum of fish parvalbumin treated with HPP. It was also observed that fluorescence intensity decreased with the increase in pressure. In a word, changes in tertiary structure can mask or expose epitopes and alter allergenicity, and the way of fluorescence chromatography can help to distinguish the changes in tertiary structure.

Besides noncovalent bond forces, the disulfide bond of covalent force is integral to maintaining conformation. For arginine kinase of crustaceans, a proven allergen, Han et al. [62] found its secondary and tertiary structure would be loose due to the breakage or formation of the disulfide bond, giving rise to potential allergenicity. The free sulfhydryl content (FSC) is related to breakage or formation of disulfide bond (S-S). Therefore, to a certain degree, FSC can explain the change in 3D structure, including conformational expansion and polymerization [63]. Conformational changes lead to shaping in epitopes, which affect allergenicity. Furthermore, studies have shown that targeted destruction of disulfide bonds can reduce the IgE-binding capacities of food allergens [64]. Under the various effect of non-thermal processes (e.g., oxidation, extrusion, vibration, and cavitation), molecules may aggregate, and free sulfhydryl groups would bind to each other to form disulfide bonds, resulting in a new conformation [11,45,50]. For instance, the FSC of allergen of prawn decreased significantly after CP treatment [28]. This may be caused by the formation of new intramolecular/intermolecular disulfide bonds. In addition, some studies suggested that the oxidation effect may expose the internal sulfhydryl groups and reduce the steric hindrance of producing disulfide bonds [54]. Similar results occurred when allergens were treated with HPP, but it was closely related to the treated pressure. It was found that high pressure treatment (200~400 MPa) could reduce FSC in allergens, but at the higher pressures (500~600 MPa), FSC was higher than it was at low pressure [27,65]. This result indicated that high pressure further changed protein conformation [37]. Accordingly, the stability of disulfide bond would shape the allergenic potential of seafood allergens, and it could be modified under the non-thermal processing treatment, forming a new conformation.

## 3. Analysis of Allergenic Changes of Seafood Allergens Induced by Non-Thermal Processing

### 3.1. IgE-Binding Capacities/Immunoreactivity Analysis

Seafood allergy is a type I allergic reaction mediated by IgE. The ability of seafood allergens to combine with human serum is the most direct indicator of immune response, namely, its IgE-binding capacities (antigenic integrity). When applying animal antibodies to conduct, it is called immunoreactivity. They are the ability of IgE/IgG to bind to epitopes [13]. In vitro immunoassays, specific IgE in the serum of allergic patients is used to react with the allergens, following characterization of its IgE-binding capacities through enzyme reactions. Enzyme-linked immunosorbent assay (ELISA), western blotting, and dot blotting are the three most common methods for IgE-binding capacities/immunoreactivity analysis.

#### 3.1.1. ELISA Analysis

ELISA can quantify the IgE-binding capacities/immunoreactivity of antigens, including indirect competition ELISA (IC-ELISA), indirect ELISA, and sandwich ELISA (shown in Figure 2). Due to the false-negative defects, direct ELISA is not accurate for seafood allergens detection [66]. In indirect methods, the processed antigen are coated as a stationary phase, then the serum of patient containing the specific antibodies and the enzyme-labeled secondary antibody would add consecutively [67]. For example, Jin et al. [27] used indirect ELISA (conducted by human serum) to test the tropomyosin of squid (*Todarodes pacificus*) treated by HPP (600 MPa, 20 °C, and 20 s) and found its IgE-binding capacities decreased by 38%. It could be attributed to the unfolding of protein and modification of secondary structure. In another study, Zhang et al. [11] also used indirect ELISA to test the hemocyanin of squid (*Todarodes pacificus*) treated by HPP (600 MPa, 25 °C and 20 s) and found its IgE-binding capacities and immunoreactivity decreased by 40% and 22%, respectively. The reason seemed to be obvious that it was the response of structure of allergens to HPP. Owing to the loose structure induced by ultrasound treatment (800 W, 15 min), a 46% reduction in IgE-binding capacities of tropomyosin of shrimp (*Exopalaemon modestus*) was detected by indirect ELISA [39]. UV also reduced the IgE-binding capacity of shrimp (*Litopenaeus setiferus*) by 25% using the same detection [29]. IC-ELISA has the process that antigen compete to captured antigen, so that it has an amplification effect and high sensitivity [25]. Sandwich ELISA uses the antibody as the stationary phase, and then adds the treated sample and the enzyme-labeled antibody consecutively. In a recent study, Dong et al. [48] used sandwich ELISA to find that the content of tropomyosin of prawn (*Litopenaeus vannamei*) decreased by 76% after ultrasound treatment (400 W, 20 min). However, it was worth noting that the results obtained by sandwich ELISA and indirect ELISA were slightly different. In a study, the former method showed that IgE-binding capacities of shrimp tropomyosin reduced by 70% after CP treatment 5 min, while the latter only showed a 40% reduction [68]. This may be due to the structural change of the allergens treated by CP, which reduced their connection to the carrier. In a word, from the results of ELISA, non-thermal processing technologies have a great effect on reducing IgE-binding capacities/immunoreactivity of seafood allergens. Notably, compared to others types of ELISA, the indirect competition ELISA may equip the higher WOE.

#### 3.1.2. Western Blotting Analysis

Compared with ELISA quantitative detection, western blotting is used to qualitatively judge by imprinting depth. It contains information of molecular weight, thus possibly determining which specific protein’s allergenicity changes during non-thermal processing treatment. Although using monoclonal antibodies in ELISA can achieve this effect, the preparation process of antibodies is too cumbersome and expensive. For example, the protein blotting of fish (*Hypophthalmichthys molitrix*) treated with HPP did not change significantly, indicating that their IgE-binding capacities did not vary greatly [46]. Moreover, in another study, it can be seen that the protein blotting gradually weakened with the increase in ultrasonic intensity, indicating that its IgE-binding capacities gradually decreased. Furthermore, with the rise in ultrasound intensity, the protein blotting slipped down, and the molecular weight became smaller, indicating that the ultrasonic processing caused the degradation of allergic proteins [39]. Therefore, western blotting can show the changes in IgE-binding capacities/immunoreactivity accompanied by information of real-time molecular weight. However, the characterization of IgE-binding capacities/immunoreactivity determined by western blotting and ELISA is not necessarily consistent. For example, in a study, the blotting showed that IgE-binding capacities of allergens disappeared after ultrasound treatment at 800 W. Contradictorily, ELISA results showed the allergens still had high allergenic potency [39]. It was concluded that ultrasound treatment broke the allergens into small fragments (including some linear epitope fragments), which are too small to be captured by the gel during the electrophoresis phase. This indicates that ELISA is more sensitive than western blotting in detecting allergenic potential [69]. Summarily, with the consequences of western blotting, allergens usually degrade after non-thermal treatment, resulting in the reduction in IgE-binding capacity.

#### 3.1.3. Dot Blotting Analysis

Dot blotting is similar to western blotting, in which the depth of blotting determines IgE-binding capacities. However, dot blotting does not require the process of electrophoresis, and it can quickly test the changes in the IgE-binding capacities of the total proteins after non-thermal treatment. The reduction in IgE-binding capacities of UV on shrimp (*Litopenaeus setiferus*) extractions were verified by dot blotting [29], since without electrophoretic separation, dot blotting characterizes the changes throughout the sample after non-thermal treatment.

### 3.2. Allergenicity Analysis

The definition of allergenicity means the potential of allergens to induce allergic reactions. It should simulate the process of immune disorders in the body. Thus, the analysis of allergenicity includes in vitro cell model assays (simulating mediators release and epithelial transport), in vivo assays (skin prick tests), and animal trials. Comparing the WOE of each immunological analysis, cell model assays and animal trials have the highest WOE, IgE-binding capacity is acceptable, and immunoreactivity has the lowest WOE [13].

#### 3.2.1. Cell Model Assay Analysis

The cell model assay, namely functional biological assays, are used to evaluate the ability of the allergens to induce a variety of responses in the body, including the release of allergenic mediators and activation of primary T cells (shown in Figure 3) [70,71]. Therefore, the cell assays evaluate the allergenicity at the cellular level and the IgE-binding capacity at the molecular level. Common cell models include mast cell (MC) and basophil cell models, dendritic cell (DC) models, and intestinal epithelial cell (IEC) models [70,71,72,73,74,75].

The MC model represents the ability that allergen stimulates the MC to release allergenic mediators (histamine and β-hexosaminidase) [76]. The common MC line is HMC-1, but its expression of surface Fcε R I receptor is low. Thus, it has not been widely used. Basophil models include human cell lines (KU812 and KU812F) and rat cell lines (RBL-2H3). Moreover, the degree of their activation is reflected by the detection of CD63, histamine, tryptase (TPS), interleukin 4 (IL-4), and β-hexosaminidase (β-hex) [77]. In a study, the KU812 model was used to evaluate the allergenicity of tropomyosin of shrimp, and the results showed that the histamine release, β-hex, CD63, and IL-4 significantly decreased after processing treatment [71]. The same cell model was also used to characterize fish allergens (parvalbumin) [75]. Furthermore, Luo et al. [77] used the RBL-2H3 model to evaluate the allergenicity of parvalbumin of fish after digestion by digestive enzymes.

DCs can function as antigen uptake and delivery [70,78]. In addition, DCs are thought to determine the direction of differentiation of naïve T cells and thus modulate the immune response [79]. Although the relationship between DCs and the allergenic potential of protein is not direct or clear, DCs are responsible for presenting antigen, which is an important process in the sensitization and reaction stages. For this reason, the degree of DCs maturity is also closely related to the allergenicity of proteins [80]. Moreover, the expression level of molecules (e.g., MHC II, CD80, CD86, CD103, IL-6, and IL-2) on the surface of DCs can be used to assist in the analysis of allergenicity of proteins after non-thermal treatment [73,81,82]. For example, Wu et al. [70] used the DCs model to evaluate the allergenicity of parvalbumin of fish and measured the release of inflammatory factors (IL-6 and IL-10). The high levels of inflammatory factors also suggested that fish allergens activated the DCs cells.

The mucosal immune system is the first line of defense of the immune system, and IECs are a vital part of it, which acts as a barrier to food allergens [82]. The transmittance of allergens has also become a subsidiary index to evaluate the allergenicity of food allergens because high transmittance means that more allergens can penetrate IEC, thereby inducing more severe allergic reactions. Generally, the Caco-2 cell model is applied to simulate the transport of food allergens by human IECs [83]. Allergens undergo significant structural changes after transport by Caco-2, especially when decomposed by digestive enzymes [84]. Moreover, the allergic reaction also inhibits proliferation of Caco-2 cell and promotes secretion of cytokine [72]. Zhang et al. [85] found that the allergen of shrimp whose IgE-binding capacities decreased, as detected by ELISA, could reduce the increment of Caco-2 and inhibit the release of IL-8 (the mediator to active the basophils).

Although the immune cell models are perfected, few reports have used them to assess the effectiveness of non-thermal processing techniques on seafood allergens. Cellular models can be employed to assist in explaining the modification of allergenic protein by non-thermal processing treatment at the cellular level, which helps comprehensively assess changes on allergenicity of seafood after non-thermal treatment.

#### 3.2.2. Animal Trials Analysis

An imbalance of immune system would cause allergic reactions. It involves many immunoregulatory processes, including gastrointestinal digestion, epithelial transport, microbiota barriers, interleukin upregulation, and mediator release (shown in Figure 3). Thus, in vitro immune response alone cannot adequately reflect the reducing effect of allergenicity induced by non-thermal processing techniques. However, there is a big risk for an allergic reaction happening in the human body, as well as ethical issues. For this reason, animal models offer a great surrogate. Commonly, animal models include Balb/c mice, C57BL/6 mice, C3H/HeJ mice, and DBA2 mice [86]. They are often selected for several generations before inducing sensitization to certain foods. Specifically, the establishment of the animal model includes the sensitization and challenge phase. In the sensitization phase, mice have been sensitized to the targeted allergen by feeding small amounts of the native allergen and immune adjuvant. This phase usually lasts about a month. Afterwards, going into the challenge phase, sensitized mice are fed large amounts of the treated allergen, which would induce allergic reactions. Finally, the degree of allergic reaction in mice is determined by several indicators, including physiological response, the IgE level in the serum, histamine content, and various cytokines content [82]. The allergenic potential and oral tolerance of food allergens after non-thermal processing treatment can be identified through animal trials. In addition to external allergic symptoms, animal trials also include the content on immunoglobulin, intestine, immune cell, and cytokine. Fu et al. [87] fed Balb/c mice with tropomyosin of shrimp and found that the allergic mice lost weight and developed allergic symptoms, including diarrhea and greasy hair. Notably, the male mice did not show significant allergic reactions. With the appearance of allergic symptoms in the challenge phase, the rising levels of histamine, IgE, and IgG2a in blood indicated that allergic reactions occurred. In addition, the level of some cytokine also rose significantly in the sensitized group, such as Il-4, IL-5, and IL-13. In contrast, level of IL-17 decreased, which would affect the polarity differentiation of ILC 2 cells and generate B cells. Sections of the intestine (duodenum, jejunum, and ileum) also showed rising level of mast cells and accumulation of eosinophils after allergic elicitation [88]. Long et al. [47] also found that the challenged mice did not show allergic symptoms after feeding allergens of shrimp treated by HPP combined with heat. Moreover, compared with the control group (fed with untreated allergens), there was a lower level of histamine expression in their serum. Therefore, the reduction effect on allergenicity of seafood allergens by non-thermal processing treatment can be more directly characterized in animal trials.

#### 3.2.3. Skin Prick Tests (SPT) Analysis

SPT is used to judge whether the patient is allergic to a specific antigen in clinical diagnosis [89]. It applies the allergens on the skin and then uses a needle to pierce it into the skin. If the patient is allergic to this allergen, the allergen will bind to the antibody and stimulate mast cells to release mediators, causing red spots to appear on the skin of patient after a few minutes [90]. Afterward, the diameter of wheals is used to determine the degree of allergic reaction. Therefore, SPT can help evaluate the allergenicity of seafood allergens according to the diameter of wheals. Compared with prolonged animal trials, SPT can be done in minutes [91]. It is worth noting that although the sensitivity of skin prick is high, its specificity is low due to a series of antibodies contained in the body. Lavilla et al. [92] used SPT to determine the allergenicity of food allergen after HPP treatment but found the opposite results to ELISA. This may be related to the low specificity of SPT. However, Gamez et al. [93] adopted shrimp protein extract as the allergen and obtained consistent results between SPT and ELISA. These unexpected results emphasize the importance of in vivo assays in evaluating the effects of non-thermal processing, despite negative bias in SPT tests [89].

### 3.3. In Vitro Digestibility Analysis

Digestibility is an important index to evaluate the ability to be absorbed in the digestive tract. Generally, food allergens are taken orally, subsequently inducing an allergic reaction in the digestive tract if the proteins maintain a relatively integral structure [64,94,95,96]. In food allergy research, digestibility is usually considered the resistance to digestive enzymes (e.g., pepsin, trypsin, and chymotrypsin). It practices by determining the degree of enzymatic hydrolysis of digestive enzymes to allergens in a specified time. If the allergens have a low resistance to digestive enzymes, their amino acid sequence will be destroyed due to the greater degree of enzymatic hydrolysis. This may result in the loss of epitopes, causing the immune cells cannot to recognize them. Therefore, the allergenicity of food allergens may relate to their digestibility. It determines whether allergens can pass through the gastrointestinal tract and then cause anaphylaxis and inflammatory response. For this reason, it is gradually used as an index or a characterization in food allergens research. Moreover, it also has been recommended as a criterion for evaluating food allergenicity by some international organizations [97]. For instance, Huang et al. [98] studied the relationship between the IgE-binding capacities of tropomyosin of mud crab and its digestibility. The results revealed that tropomyosin showed resistance to pepsin but was relatively easy to degrade by trypsin and chymotrypsin. Furthermore, its allergenicity was gradually decreased with the increasing degree of enzymatic hydrolysis. Lv et al. [67] also found that the IgE-binding capacities of allergens of shrimp decreased with increasing digestion degree. Liu et al. [99] recommended that tropomyosin from Pacific white shrimp (*Litopenaeus vannamei*) was possibly more allergenic than it from Grass prawn (*Penaeus monodon*) due to its higher digestion stability as well. Therefore, the digestion ability of proteins is widely used as an indicator related to its allergenic potency in food allergen research, especially seafood allergens. Nevertheless, the results of this indicator are usually affected by many factors, including the pH, the proteolytic, and the enzyme-to-protein ratio, so it does not have a specific criterion. Furthermore, the relationship between the digestibility and allergenicity of food allergens was not fully understood. Some studies even showed no significant correlation between the digestibility of certain food proteins and their allergenicity [100], but this did not prevent it from becoming a worthy reference index in the studies of seafood allergen. In addition, several research about non-thermal processing treatment take advantage of this method to investigate the mechanism of reducing effect on allergenic potency [27,48].

It has been shown that most seafood allergens would be more tolerable to digestive enzymes than non-food allergens [97], such as tropomyosin of crab, a proven allergen, which is hardly digested by pepsin [96]. Hence, digestibility is recognized as an effective indicator of structure and allergenicity of allergen during the non-thermal processing treatment. For example, Jin et al. [27] explored the relationship between IgE-binding capacities of tropomyosin Tod p1 (TMTp1) of squid and its digestibility after HPP treatment. In particular, the digestion stability assay of simulated gastric fluid (SGF) and simulated intestinal fluid (SIF) revealed that with the digestion time increasing, the IgG/IgE special reactivity of TMTp1 decreased, and it was more resistant to pepsin than trypsin. In addition, HPP treatment promoted TMTp1 to be digested due to its structural unfolding and exposure. Coincidentally, Dong et al. [48] also reported that the digestibility of proteins of shrimp increased from 76.42% to 83.95% after 20 min 400 W of ultrasound processing, while the content of tropomyosin (chief allergen) decreased by 76%. These results could help explain the structural modification of ultrasound treatment, including a decrease in α-helix content and an increase in β-sheet. On the other hand, the digestion stability of food proteins is also a critical food property, directly relating to edibility after non-thermal processing. Therefore, digestibility of seafood allergens can not only reflect the allergenicity, but also contribute to explaining the structural changes induced by non-thermal processing technology. It is a worthwhile and valuable experimental indicator in food allergens research.

### 3.4. Gut Microbiota Analysis

In addition to well-established serum assays and macroscopic animal trials, changes in gut microbes may also provide a new way to characterize food allergic reactions. As an important line of defense against food allergy, the barrier of gut microbiota located in intestine is closely related to the immune system. Some studies have shown that the composition of gut bacteria may promote or suppress food allergies (shown in Figure 4) [101]. The mucosal layer of the intestine is home to a large number of microorganisms (gut microbiota), which play a significant role in regulating the immune system and maintaining the intestinal barrier (including mechanical, biological, chemical, and immune barriers) [102]. It was shown that intestinal microbes could affect food allergic reactions by altering the permeability of intestinal mucosal and state of immune system [103,104]. For example, certain gut bacteria (e.g., *Lactobacillus acidophilus*) can enhance function of IECs barrier by strengthening their tight junctions. Moreover, short-chain fatty acids produced by bacterial fermentation can also promote the proliferation and differentiation of IECs [105]. Besides playing a role of barrier, the composition of microbiome influences human sensitization to food allergens [106]. It could affect the direction of differentiation of immune cell [107]. For example, the number of *Bacteroidetes* decreased and *Firmicutes* increased after a shrimp-stimulated allergic state [87], which would promote the differentiation of ILC2 into Th2 and regulate Treg-mediated immune tolerance [108]. Moreover, *Lachnospiraceae*, a family in the Clostridia class, also increased significantly, which was associated with oral tolerance [109]. Feehley et al. [110] also pointed out that the Clostridia class affected the food allergic reactions by regulating the level of Treg cells. Similarly, Wu et al. [70] demonstrated that parvalbumin of fish also caused disturbance of intestinal microbiota. At present, although the quantitative and qualitative relationship between them is still unclear, a large number of studies have evidenced that the occurrence of anaphylaxis is inevitably accompanied by significant changes in the composition of intestinal flora [82,111,112]. As an essential part of the body’s immune system and metabolic function, intestinal microbes can be used to characterize the treatment effect and potential safety problems of non-thermal processing more comprehensively.

## 4. Conclusions and Future Trends

A large number of studies have confirmed that non-thermal processing techniques can reduce the IgE-binding capacities of seafood allergens. However, due to its fuzzy processing mechanism, such a conclusion needs more robust, thorough, and validated assessments to verify it. From the perspective of structure and allergenicity, non-thermal processing would modify the structural integrity of seafood allergens, leading to a reduction in IgE-binding capacities/immunoreactivity. After gathering the available reported research, non-thermal processing techniques had the potential to break down the covalent/non-covalent bond of seafood allergens; through the structural analysis, allergic proteins would undergo degradation, aggregation, uncoiling, unfolding, folding, and exposure behaviors, resulting in a significant change of conformation. Subsequently, these disruptions mask or expose the epitopes, influencing their allergenic potency. Furthermore, structural incompleteness induced by non-thermal processing treatment would alter their resistance to digestive enzymes, especially pepsin. In addition, functional assays, in vivo assays, or animal trials should be conducted to evaluate the allergenicity of treated seafood allergens. Only in this way could the actual reducing effect on allergenicity induced by non-thermal processing be revealed.

In order to use non-thermal processing technologies as new processing ways, it is necessary to use various methods to analyze their processing effect and processing mechanism, which would test their effectiveness, reliability, and safety. The relationship between structure and allergenicity of seafood allergen needs more data and studies to support it. In addition, some other methods of evaluation related to allergenicity need to be perfected, such as Ca^2+^ binding ability, the interaction of various cytokines, and X-ray diffraction. Only after a comprehensive analysis on the effect of non-thermal processing technologies can they provide practical support for food industries in the future.

## Figures and Tables

**Figure 1 molecules-27-05857-f001:**
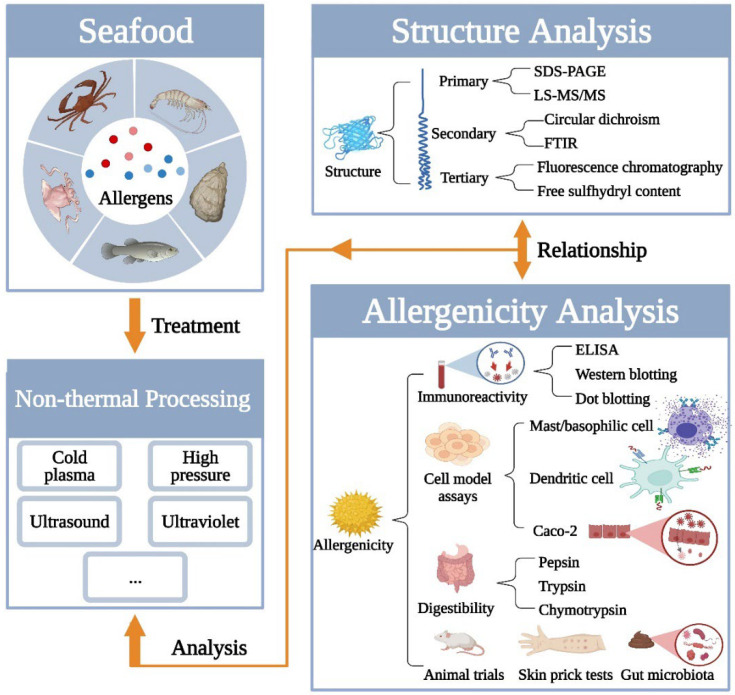
Analysis for structure and allergenicity changes of seafood allergens induced by non-thermal processing.

**Figure 2 molecules-27-05857-f002:**
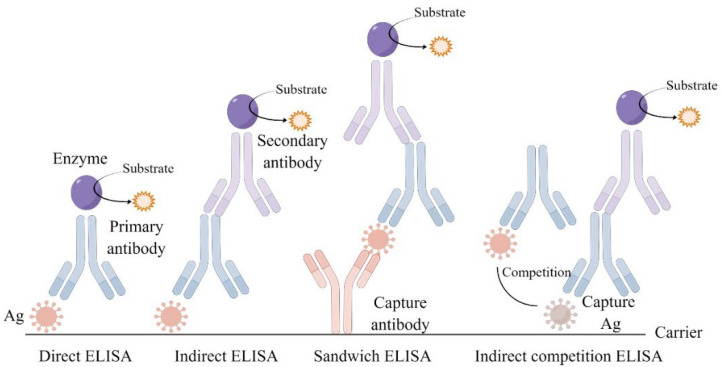
The type of enzyme-linked immunosorbent assay (ELISA). Note: Ag: Antigen.

**Figure 3 molecules-27-05857-f003:**
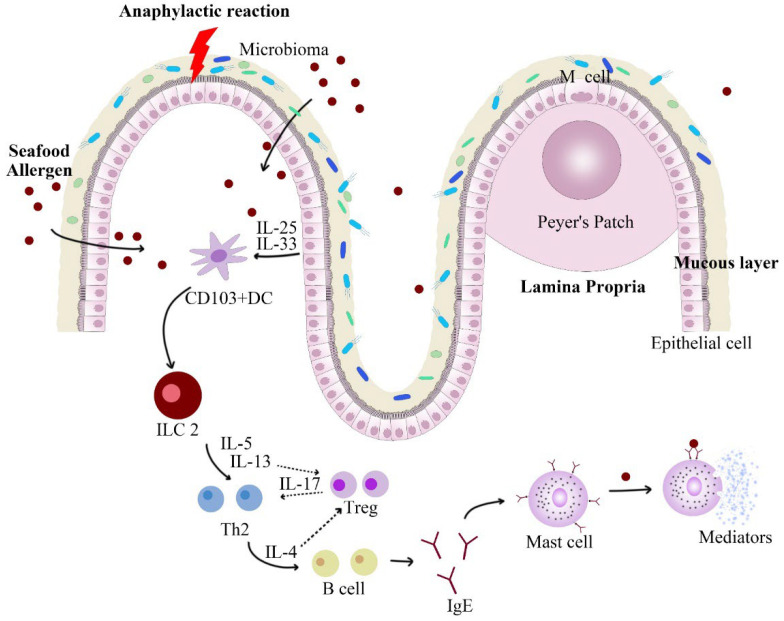
Main mechanisms by which the breakdown of tolerance to food antigens can occur. When food allergens stimulate the mucosal system, the damaged epithelial cells will secrete cytokines, such as IL-33, to induce the differentiation of ILC 2 into Th 2, and secrete related cytokines, such as IL-13. Il-4 inhibits the differentiation of ILC 2 into Treg, ultimately leading to the production of allergen-specific IgE. IgE binds to Fcε R I fragments on mast cells to activate mast cell release mediators and induce food allergy symptoms.

**Figure 4 molecules-27-05857-f004:**
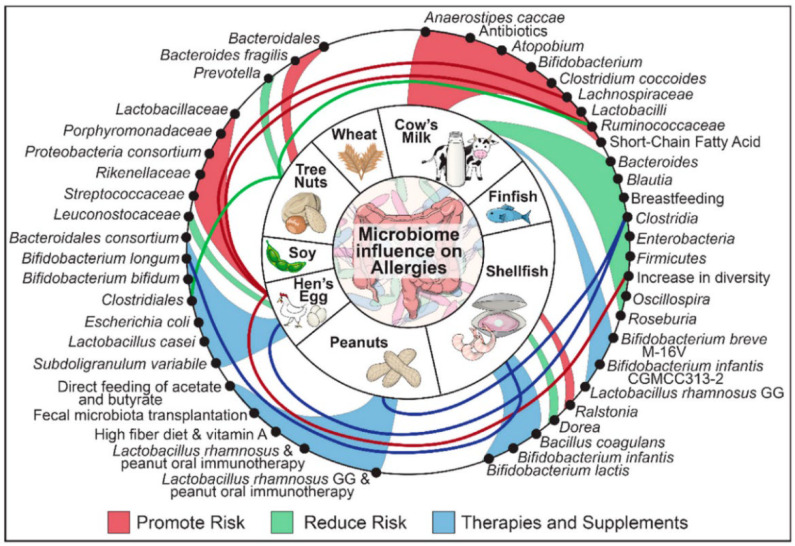
The relationship between the big eight food allergens and gut microbiota [113]. The advantage of *Ralstonia* would promote the allergic risk of shellfish, while *Dorea* would reduce the risk. Moreover, *Bacillus coagulans*, *Bifidobacterium infantis*, and *Bifidobacterium lactis* have the potential for shellfish allergic reactions therapies.

**Table 1 molecules-27-05857-t001:** An overview of studies evaluating the effects of non-thermal processing on the structure of seafood allergens.

Non-Thermal Techniques	Target Allergen	Food Material	Treatment Conditions	Evaluations	Results	IgE-Binding Capacities/Immunoreactivity	References
High pressure processing (HPP)	Tropomyosin	Squid	400 MPa, 20 °C, 10 min	SDS-PAGE	No changes	Decrease	[27]
				CD	α-helix decrease, β-sheet increase		
				FSC	Decrease		
				SH	Increase		
	Hemocyanin	Squid	200–600 MPa, 25 °C, 20 min	SDS-PAGE	Band intensities decrease	Decrease	[11]
				FTIR	α-helix decrease, β-sheet increase		
				FSC	Decrease		
				SH	Increase		
				SAXS	R_g_ decreased, 〈N_agg_〉_G_ and 〈N_agg_〉_Q_ increased		
	NS	Squid	200–600 MPa, 20 min	SDS-PAGE	Band intensities decrease	NS	[33]
	NS	Fish	200 MPa, 25 °C, 20 min	SDS-PAGE	New bands at 95 and 26 kDa	Decrease	[44]
				FSC	Decrease		
				SH	Increase		
	Parvalbumin	Fish	300–600 MPa, 10 min	CD	α-helix decrease	NS	[45]
				FSC	Decrease		
				SH	Increase		
				Laser Raman spectroscopy	α-helix decrease, β-sheet increase		
	Actomyosin	Fish	200–600 MPa, 10–50 min	SDS-PAGE	Band intensities decrease	NS	[30]
				FSC	Decrease		
				SH	Increase		
	NS	Fish	100–300 MPa, 20 °C, 10–60 min	SDS-PAGE	No changes	No changes	[46]
				CD	Structure changes		
	Tropomyosin	Shrimp	100–600MPa, 0–30 min	/	/	Decrease	[47]
Cold plasma (CP)	Tropomyosin	Shrimp	3–15 min	SDS-PAGE	No changes with DTT, slightly fade without DTT	Decrease	[28]
				CD	α-helix decrease, β-sheet increase		
				FSC	Decrease		
				SH	Increase		
	Actomyosin	Shrimp	1–5 min	SDS-PAGE	No changes	NS	[31]
				CD	α-helix decrease, β-turns increase		
				FSC	Decrease		
				SH	Increase		
Ultrasound	Total proteins	Shrimp	400 W, 20 min	SDS-PAGE	Band intensities decrease	Decrease	[48]
				FTIR	α-helix increase, β-sheet increase, β-turns decrease		
	Tropomyosin	Shrimp	800 W, 30–180 min	SDS-PAGE	Band intensities decrease	Decrease	[49]
	Tropomyosin	Shrimp	100–800 W, 15 min	SDS-PAGE	Band intensities decrease	Decrease	[39]
				CD	α-helix decrease, β-sheet and β-turns increase		
				FSC	Decrease		
				SH	Increase		
	NS	Fish	200–950 W, 60 min	SDS-PAGE	No changes	NS	[50]
				FSC	Decrease		
				SH	Increase		
				FS	Blue shifts		
Ultraviolet (UV)	Tropomyosin	Shrimp	4 min	SDS-PAGE	Band intensities decrease	Decrease	[51]
	Tropomyosin	Shrimp	0–6 min	SDS-PAGE	Band intensities decrease	Decrease	[29]

## Data Availability

Not applicable.

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
