# Peer review of "Comprehensive Analysis of the Structure and Allergenicity Changes of Seafood Allergens Induced by Non-Thermal Processing: A Review"

_molecules, 2022, doi:10.3390/molecules27185857_

Round 1

Reviewer 1 Report

    In this review, Wang et al. described the current knowledge of non-thermal processing on the allergic potential of seafood. The topic is timely, as a lot of studies currently assessed the impact of novel processing technologies on allergenicity. Overall, the review is quite exhaustive, but had a few mistakes need to revise. 

Comments:

Line14 - ‘comprehensive analysis to the allergenicity changes’ instead of ‘comprehensive analysis of the allergenicity changes’

Line 20 - ‘not convincing enough’ instead of ‘not enough convincing’

Line 27 - popular instead of famous

Line 43 - ‘food-allergy patients’ instead of ‘food-allergic patients’

Line 45 - Why is heat treatment is a useful method to tackle seafood allergen. Please add. 

Line 50 - ‘on sensory properties and nutritional value of food’ instead of ‘of food sensory properties and nutritional value’

Line 51- ‘ineffective’ is too absolute

Line 61 - ‘reducing’ instead of ‘reduction’

Line 79 - squid tropomyosin

Line 80-81 - Please supply the duration of HPP

Line 85 - ‘unclear’ instead of ‘uncertain’

Line 86 - ‘In addition,’ instead of ‘on the other hand’ 

Section "Cell model assay analysis" - the authors describe the potential to study the effect of non-thermal modification on protein allergenicity, but they do not provide any concrete example of study. Are there any examples in current study?

Figure 4 - the figure should be more specific for the case of seafood allergens. Please revise.

Reviewer 2 Report

The manuscript entitled “Comprehensive analysis of the structure and allergenicity changes of seafood allergens induced by non-thermal processing: a review” described the effects of non-thermal treatment on structure and allergenicity of seafood allergens and corresponding characterizing methods. In general, this review is well done from the perspective of integrity and novelty, especially the session of in vivo analysis. However, some following modifications are needed.

Comments:

Line 79 - squid tropomyosin. There are some similar usages in the article. Please check throughout the article.

Line 131- I don’t think it is proper to put “After” here.

Line 135 - I don’t understand what the word “closed to” mean here.

Line 139 - “moved” should be replaced by “move”. This is supposed to be active, not passive.

Lines 157-158 - “may attribute to” should be replaced by “may be attributed to”.

Line 162 - “merely” may be more suitable than “only”

Line 176 - “allergen” should be replaced by “allergens”.

Line 186 - “changes” should be replaced by “changed”.

Line 189 - “allergen” should be replaced by “allergens”. Please check throughout the article.

Line 195 - “transform in” should be replaced by “transform of” or “transform from”.

Line 201-201 - “who applied high-intensity ultrasound processed proteins of shrimp” seems weird. “Who applied high-intensity ultrasound process proteins of shrimp” seems better.

Line 216 - “Due to” is not proper when followed by a complete sentence.

Line 289 - “IgE-binding capacity” instead of “allergenicity”

Line 382 and 384 - “allergenicity of allergens” seem redundant. Please correct them.

Line 420 - “was” should be replaced by “is”.

Figure 3- I think Figure 3 is not needed.

Apart from the mistakes mentioned above, some similar mistakes about grammars should be checked again.

Reviewer 3 Report

A few minor comments that require author's attention (mainly language and grammatical mistakes). These suggestions will help to improve the quality manuscript. Please revise carefully.

Line 44 heated change to hot

Line 52 seafood allergens

Figure 1 structure-allergenicity changes change to structure and allergenicity changes

Line 84 analyses change to analysis

Line 292-294 Please revise the sentence

Line 466 immune cells

Line 481 food allergen research
